# Aptameric hirudins as selective and reversible EXosite-ACTive site (EXACT) inhibitors

Haixiang Yu[1], Shekhar Kumar[2], James W. Frederiksen[1], Vladimir N. Kolyadko[2], George Pitoc[1], Juliana Layzer[1], Amy Yan[1], Rachel Rempel[1], Samuel Francis[3], Sriram Krishnaswamy[2] ✉ & Bruce A. Sullenger[1,4] ✉

Potent and selective inhibition of the structurally homologous proteases of coagulation poses challenges for drug development. Hematophagous organisms frequently accomplish this by fashioning peptide inhibitors combining exosite and active site binding motifs. Inspired by this biological strategy, we create several EXACT inhibitors targeting thrombin and factor Xa de novo by linking EXosite-binding aptamers with small molecule ACTive site inhibitors. The aptamer component within the EXACT inhibitor (1) synergizes with and enhances the potency of small-molecule active site inhibitors by many hundred-fold (2) can redirect an active site inhibitor's selectivity towards a different protease, and (3) enable efficient reversal of inhibition by an antidote that disrupts bivalent binding. One EXACT inhibitor, HD22-7A-DAB, demonstrates extraordinary anticoagulation activity, exhibiting great potential as a potent, rapid onset anticoagulant to support cardiovascular surgeries. Using this generalizable molecular engineering strategy, selective, potent, and rapidly reversible EXACT inhibitors can be created against many enzymes through simple oligonucleotide conjugation for numerous research and therapeutic applications.

Hematophagous organisms, such as leeches, ticks, mosquitos and nematodes, use potent inhibitors of the coagulation proteases to acquire blood meals[1]. These inhibitors are frequently peptides that engage the target protease through exosite and active site interactions to fulfill the biological needs of high potency and rapid onset of action during feeding[2–9]. This strategy is exemplified by hirudin from the medicinal leech that targets thrombin using an N-terminal active site binding motif linked to a C-terminal region that binds anion binding exosite I (ABE1) (Fig. 1a)[2]. The two motifs contribute synergistically to binding and achieve a $10^{-11}$M inhibition constant ($K_i$) which is greatly affected when binding by either domain is impaired[10,11]. Tick anticoagulant peptide (TAP) secreted by the *Ornithodoros moubata*[7] also achieves potent inhibition

of factor Xa by simultaneously binding the active site and an extended surface on the protease, and inhibits free factor Xa ($K_i = 0.18$ nM) and complexed factor Xa within prothrombinase ($K_i = 5.3$ pM) to achieve potent anticoagulation through an analogous mechanism[9,12]. The evolution of such bivalent inhibitors by different blood-feeding organisms inspired us to pursue a synthetic but analogous bifunctional engineering strategy using a small molecule active site inhibitor tethered to an exosite-binding aptamer.

In this work, by evaluating the binding and inhibitory mechanisms of such EXosite and ACTive site (EXACT) inhibitors, we show that the exosite-binding aptamer can be utilized to modulate the apparent binding affinity and target selectivity of the small molecule moiety.

[1]Department of Surgery, Duke University, Durham, NC, USA. [2]Research Institute, Children's Hospital of Philadelphia, Philadelphia, PA, USA. [3]Department of Emergency Medicine, Duke University Hospital, Durham, NC, USA. [4]Departments of Pharmacology & Cancer Biology and Biomedical Engineering, Duke University, Durham, NC, USA. ✉e-mail: skrishna@pennmedicine.upenn.edu; bruce.sullenger@duke.edu

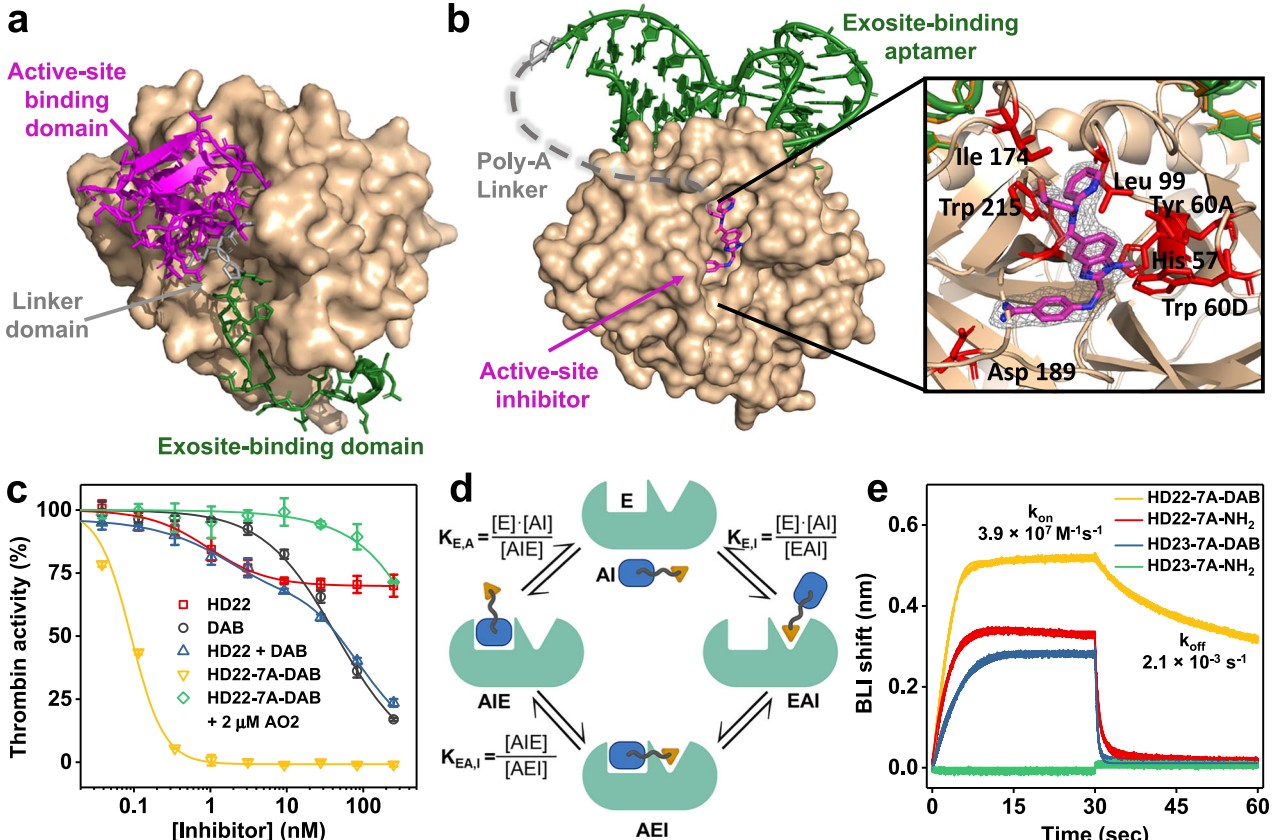

**Fig. 1 | Rational design of thrombin-binding EXACT inhibitor. a, b** Crystal structures of hirudin-thrombin complex (**a**) in comparison to HD22-7A-DAB-thrombin complex (**b**). In both inhibitors, the active site-binding domain (magenta) and the exosite binding domain (green) are connected by a linker domain (gray), which allows synergistic binding to thrombin (tan). In **b**, the active site was zoomed in to show thrombin's interacting residues (red) with dabigatran. The composite omit 2Fo·Fc map contoured at 1.0 σ shows the electron density for DAB as gray mesh. **c** Thrombin inhibition by HD22-7A-DAB and other inhibitors in fluorescent peptidyl substrate cleavage assay. HD22-7A-DAB showed a significantly higher inhibitory effect of thrombin than DAB, HD22, or their equimolar mixture, and can be efficiently reversed by an antisense oligonucleotide (AO2). Data are presented as mean values +/- SD calculated from three independent experiments (*n* = 3). **d** two-step binding model of EXACT inhibitor (AI) to protease (E) determined by dissociation constants $K_{E,A}$, $K_{E,I}$, and $K_{EA,I}$. **e** Characterization of binding kinetics of biotinylated HD22-7A-DAB and its derivatives to thrombin (1.6 nM) by bio-layer interferometry. The illustration in panel (**d**) was created using Biorender.com.

This property can be used to manipulate the potency, specificity, and antidote-mediated reversibility of the EXACT inhibitors. This bioinspired EXACT inhibitor strategy is generalizable to different aptamer–small molecule combinations and thus represents a molecular engineering approach for generating potent, selective, and reversible inhibitors for a wide range of enzymes that are vital for health and dysregulated in disease.

## Results

### De novo rational design of an EXACT thrombin inhibitor

We investigated whether an EXACT inhibitor against thrombin could be rationally created de novo by chemically conjugating the direct thrombin inhibitor dabigatran (DAB, $K_i$ = 4.5 nM)[13] to the DNA aptamer HD22[14] (Supplementary Fig. 1 and Supplementary Table 1). HD22 binds to anion binding exosite 2 (ABE2) of thrombin with high affinity ($K_D$ = 0.5 nM) and high selectivity, but has weak anticoagulant activity as it does not bind either ABE1—the fibrinogen binding site—or the active site[14]. The crystal structure of the HD22-thrombin complex indicates that the distance between HD22's 5′ terminus and thrombin's active site is ~40 Å[15]. Therefore, we extended HD22's 5′ terminus with a poly-(7)-deoxyadenosine linker (estimated length of 49 Å)[16] to facilitate the simultaneous binding of both the HD22 and the DAB moieties. DAB's carboxyl group which is not directly involved in binding thrombin[15] was then conjugated to the 5′ amino modifier at the terminal of the linker. The resulting EXACT inhibitor is named HD22-7A-DAB (Supplementary Table 1).

The x-ray structure of thrombin complexed with HD22-7A-DAB at 2.2 Å resolution (Fig. 1b, Supplementary Fig. 2a, and Supplementary Table 2) confirms the bivalent binding mechanism of HD22-7A-DAB. The diffraction data allows unambiguous placement of both aptamer and DAB moieties onto thrombin. The aptamer-thrombin interface in HD22-7A-DAB corresponds well to the previously determined structure of HD22 bound to thrombin[15]; thus no obvious conformational changes in the aptamer or thrombin are observed with the linker and DAB added onto the aptamer (Supplementary Fig. 2b). DAB binds the S1 specificity pocket where its amidine group is appropriately juxtaposed with Asp189, as found in the previously reported structure of the thrombin-DAB complex[13]. The carboxyl group of DAB moiety is displaced from the reported thrombin-dabigatran complex possibly to accommodate the linker's position (Supplementary Fig. 2b). Only one deoxyadenine of the 7 nt linker at the 5′ terminus of HD22 is unambiguously resolved in the structure. At lower map contour levels, a partial second nucleotide is resolved, as is density potentially representing other nucleotides in the linker. These observations imply high flexibility of the linker, which presumably facilitates bivalent binding. A 1:1 binding stoichiometry of HD22-7A-DAB to thrombin was then confirmed by size exclusion chromatography (Supplementary Fig. 3).

## HD22-7A-DAB demonstrates extraordinary thrombin-binding affinity and inhibitory potency via synergistic binding

Initial velocity studies of fluorescent peptidyl substrate cleavage by thrombin were then used to assess protease inhibition by the EXACT inhibitor (Fig. 1c). DAB alone inhibited thrombin in this assay with an $IC_{50}$ (the concentration of inhibitor yielding 50% of the maximum inhibition, (Supplementary Note 1) of 50 nM), a value similar to that previously reported[17]. On the other hand, exosite-binding HD22 does not directly block the active site and despite having a low $IC_{50}$ of 0.95 nM, it can only inhibit thrombin activity by 30% at maximum. Strikingly, HD22-7A-DAB inhibited thrombin with an $IC_{50}$ of 0.1 nM, indicating a potency 500-fold higher than DAB and the aptamer-small molecule conjugate inhibited >99% of thrombin activity at a concentration of 1 nM (Fig. 1c). The marked inhibitory potency of the EXACT inhibitor very likely results from the synergistic binding of thrombin by its HD22 and DAB moieties, since an equimolar mixture of free HD22 and free DAB achieved only the additive inhibitory effects of those two molecules (Fig. 1c). We controlled for non-specific aptamer effects by generating a variant of the EXACT inhibitor whose aptamer moiety had a point mutation that abrogates its high-affinity binding to thrombin[18]. This mutated EXACT inhibitor derivative, HD23-7A-DAB, has similar potency as DAB alone, confirming that exosite binding by the aptamer is required for potent thrombin inhibition by the parental EXACT inhibitor (Supplementary Fig. 4). Similarly, replacing the DAB moiety of HD22-7A-DAB with a non-binding amine group (HD22-7A-NH2) also yielded a derivative that did not significantly inhibit thrombin.

The increased potency of the EXACT inhibitor can be explained by a two-step thermodynamic binding model (Fig. 1d and Supplementary Note 2). In the first step, either the aptamer or small molecule moiety binds to the target protease with a dissociation constant presumably similar to that of the free aptamer or small molecule. This interaction positions the other moiety in proximity to its binding site, thus facilitating synergistic binding. Notably, the second binding step is a unimolecular process and is independent of the EXACT inhibitor's concentration; this rationally engineered, juxtaposition allows effective inhibition of the protease even at a very low inhibitor concentration. This binding model is supported by binding affinity and kinetic measurements via biolayer interferometry (BLI). EXACT inhibitor HD22-7A-DAB has a $k_{on}$ of $3.9 \times 10^7\ \mathrm{M^{-1}s^{-1}}$ and a $k_{off}$ of $2.1 \times 10^{-3}\ \mathrm{s^{-1}}$ for thrombin binding, yielding a $K_D$ of 54 pM (Fig. 1e and Supplementary Fig. 5). HD22-7A-DAB shows significantly higher $k_{on}$, lower $k_{off}$, and over 100-fold lower $K_D$ compared with its monovalent derivatives (HD22-7A-NH$_2$ and HD23-7A-DAB), indicating strong synergistic binding through two domains. As a control, when both binding moieties are altered (HD23-7A-NH$_2$), no binding was observed (Fig. 1e and Supplementary Fig. 5).

## The linker length determines synergy and inhibition potency of the EXACT inhibitor

According to the two-step binding model shown in Fig. 1d, the unitless dissociation constant of the second step, $K_{EA,I}$ dictates EXACT inhibitor's binding synergy. When $K_{EA,I}$ decreases, the composition of bivalently bound complex (AEI) in all bound complexes (AIE, EAI, AEI) increases, resulting in lower $IC_{50}$ and higher maximal inhibition (Fig. 2a–c). We postulate that the linker domain regulates the $K_{EA,I}$ of an EXACT inhibitor and the potency of the EXACT inhibitor can be fine-tuned by varying its linker length. To investigate that possibility, we synthesized a series of HD22-DAB conjugates with linker lengths ranging from 0 to 30 nucleotides (nt) and tested their potency in the fluorescent peptidyl substrate cleavage assay (Fig. 2d). The inhibition pattern correlates well with our predicted binding model (Fig. 2b). Even the conjugate with the shortest linker domain (0 Å, estimated length = 8 Å) showed a high thrombin-binding affinity ($IC_{50} = 0.55 \pm 0.04$ nM), probably due to the high affinity of the HD22

moiety[14]. However, a fitted $K_{EA,I}$ of 1.8 indicated that only 35% of the conjugate binds bivalently (Fig. 2a and e). This low level is probably due to the short linker creating steric hindrance for bivalent binding. As a result, the conjugate only maximally reduced thrombin activity by 55% (Fig. 2f). With increasing length of the linker, $K_{EA,I}$ decreases as the flexibility of the linker better accommodates bivalent-binding. As expected, the $IC_{50}$ of the conjugate decreased and the maximum inhibition increased, indicating more effective inhibition of thrombin. When the linker length reaches 7 nt (estimated length 49 Å), the conjugate can fully inhibit thrombin (inhibition max > 99%) with an $IC_{50}$ of 0.10 nM. A $K_{EA,I} \sim 0.01$ indicated the aptamer and small molecule moiety almost always bind simultaneously (Fig. 2a, d–f). Further increasing of linker length to 30 nt (estimate length 18.5 nm) does not significantly alter the maximum inhibition, $IC_{50}$ or $K_{EA,I}$ of the conjugate. Such results show that the strong synergy between the small molecule and the aptamer occurs over a broad range of linker lengths as long as the spacer can span than the distance between the active site and exosite, a critical consideration for designing additional EXACT inhibitors.

## The exosite-binding aptamer domains regulates the selectivity of EXACT inhibitors

The ability of an exosite-binding aptamer HD22 to promote binding of a small molecule DAB to thrombin's active site prompted us to test if another exosite-binding aptamer can be used to manipulate the inhibition potency and selectivity of this small molecule inhibitor. It is well known that DAB is a fairly selective thrombin active site inhibitor, although it also binds factor Xa, but with over 800-fold weaker affinity ($K_i = 3760$ nM)[13]. We therefore conjugated DAB to the 5' end of a 36-nt factor Xa RNA aptamer 11F7t via a 20 nucleotide poly 2' O-methyl A linker (Supplementary Table S1)[19]. The crystal structure of FXa complexed with 11F7t indicates that a linker of that length would readily span the distance between the 5' terminus of the aptamer and the active site of FXa[20]. This EXACT inhibitor, termed 11F7t-20A-DAB, was evaluated in the fluorescent peptidyl substrate cleavage assay along with several control derivatives. As expected, DAB only weakly inhibits factor Xa ($IC_{50} > 500$ nM) (Fig. 2g). The aptamer 11F7t alone, although having high affinity to factor Xa ($K_D = 1.26$ nM), also showed no inhibition as it does not impede access to the active site of the protease (Supplementary Fig. 6)[19]. However, 11F7t-20A-DAB achieved an $IC_{50}$ 0.18 nM, demonstrating > 2500-fold enhancement of potency against factor Xa compared to free DAB. In contrast, the thrombin inhibition profile of 11F7t-20A-DAB ($IC_{50} = 120$ nM) is similar to that of free DAB and dramatically inferior to that achieved for factor Xa inhibition (Fig. 2h). This result highlights the ability of the exosite targeting aptamer domain to redirect selectivity of a small molecule inhibitor towards a different enzyme. The tight and specific binding of 11F7t to an exosite on factor Xa apparently confines the DAB moiety into close proximity with factor Xa's catalytic center, generating a high local concentration, which results in potent inhibition. Then as expected, when the exosite-binding factor Xa aptamer was replaced with the thrombin-binding aptamer HD22 (HD22-20A-DAB), the EXACT inhibitor showed an enhanced specificity to thrombin ($IC_{50} = 0.1$ nM) over factor Xa ($IC_{50} > 500$ nM) (Fig. 2g, h).

Remarkably, because the active-site inhibitor binding is a concentration-independent unimolecular process after aptamer binding, the potency of the EXACT inhibitor is insensitive to the affinity of the active-site inhibitor. For example, when the weak-binding DAB moiety of 11F7t-20A-DAB is replaced with a strong-binding factor Xa active site inhibitor apixaban-COOH (APX, $IC_{50} = 5.5$ nM), the resulting 11F7t-20A-APX exhibits almost identical potency against factor Xa ($IC_{50} = 0.17$ nM) as 11F7t-20A-DAB ($IC_{50} = 0.18$ nM) (Fig. 2i, Supplementary Fig. 7 and Supplementary Table 1). These results clearly illustrate the versatility of the exosite targeting aptamer, allowing generation of selective and potent EXACT inhibitor regardless of the

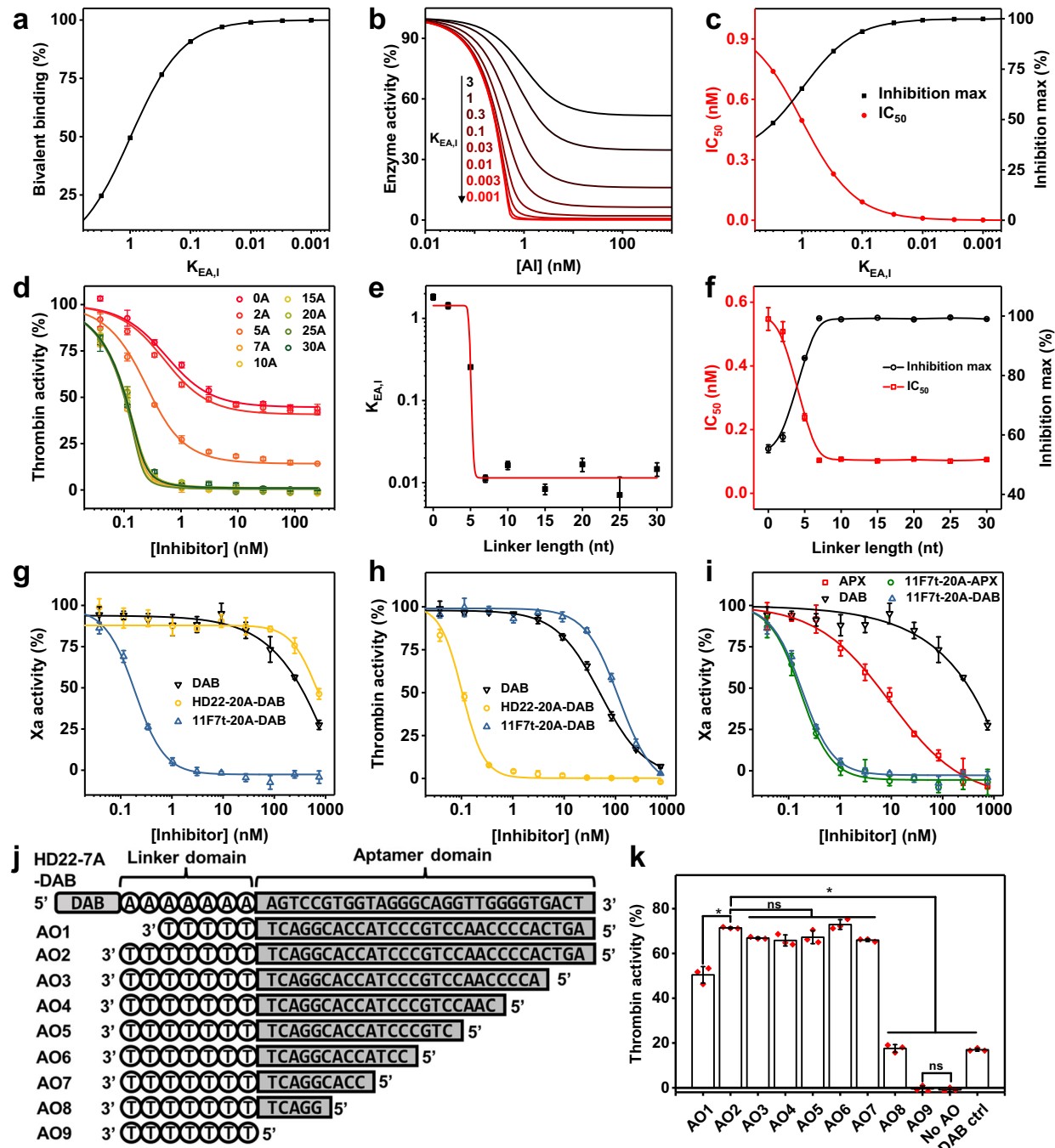

**Fig. 2 | Regulation of inhibitory properties of EXACT-inhibitors. a–c** Simulation of composition of bivalent-binding for all binding complexes (**a**) enzyme inhibition curves (**b**) and the maximum inhibition and IC$_{50}$ (**c**) with varying $K_{EA,I}$ by EXACT inhibitors with different $K_{EA,I}$ based on the proposed two-step binding model with additional assumptions (see Supplementary Note 2) **d–f** Synergistic-binding of EXACT inhibitors can be tuned by varying linker length. The inhibition curves of HD22-DAB conjugates with different dA linker lengths in fluorescent peptidyl substrate cleavage assay (**d**) indicate that the linker length regulates the $K_{EA,I}$ (**e**), IC$_{50}$ and inhibition max (**f**) of HD22-DAB conjugates. **g–i**, Exosite-binding aptamers regulate the selectivity of EXACT inhibitors. Inhibition of factor Xa (**g**) and thrombin (**h**) activity by EXACT inhibitors constructed with thrombin (HD22) or factor Xa (11F7t)-binding aptamer, respectively, demonstrate that the selectivity of the small molecule inhibitor DAB can be dictated by the exosite-binding aptamer to which it is conjugated. The inhibition potency of EXACT inhibitors is not sensitive to the affinity of small molecule active site inhibitors (**i**). **j, k** The linker domain

modulates the antidote-mediated reversibility of EXACT inhibitors. **j** Sequence of HD22-7A-DAB and different antidotes. **k** Thrombin activity (%) of each sample is measured by fluorogenic activity assay and normalized with the thrombin activity in the absence of inhibitor as 100%. Samples AO1-5 show thrombin activity in the presence of 250 nM HD22-7A-DAB and 2 μM of different AOs. Sample No AO and DAB ctrl show thrombin activity in the presence of 250 nM HD22-7A-DAB and DAB, respectively. In panels (**d**, **g–i** and **k**) data are presented as mean values +/– SD calculated from three independent experiments ($n = 3$). In panels e and f, data are presented as mean values +/– SE fitted from the data in panel d. In penal (**k**) One-way ANOVA test followed by Bonferroni's multiple comparisons test (two-sided) was used to compare between two sets of data. *$P < 0.05$, ns, not significant. Exact $P$-values are provided as follows. AO2 vs AO1, AO2 vs AO8, AO2 vs AO9, AO2 vs No AO, AO2 vs DAB ctrl: $p < 0.0001$. AO2 vs AO3: $p = 0.4144$; AO2 vs AO4: $p = 0.0807$; AO2 vs AO5: $p = 0.7064$; AO2 vs AO7: $p = 0.0984$; AO2 vs AO6 and AO9 vs No AO: $p > 0.9999$.

inherent selectivity and potency of the small molecule active-site inhibitor.

## The linker domain impacts the antidote-mediated reversibility of EXACT inhibitors

One of the valuable traits of aptamer-based inhibitors is that they can be reversed by complementary antidote oligonucleotides (AO) that disrupt the folded structure of the aptamer[21]. We next investigated if the activity of the aptamer-based EXACT inhibitor HD22-7A-DAB can be reversed using an AO. Strikingly, an AO termed AO2 (Fig. 2j) that is complementary to the entire length of the aptamer and the linker region reduced the anticoagulant potency of HD22-7A-DAB by more than 10,000-fold (Fig. 1c), despite HD22-7A-DAB's high affinity for thrombin. Notably, although AO2 does not directly interact with the DAB moiety, the antidote reduces the potency of the EXACT conjugate to an extent that is significantly lower than free DAB (Fig. 2k). This result illustrates the fundamental role of the exosite-binding component of the EXACT inhibitor in driving affinity for active site inhibition. In line with the importance of the worm-like flexible behavior for the single stranded DNA linker, hybridization of the AO to the entire length of the aptamer plus linker not only disrupts exosite-dependent binding to thrombin but stiffens the linker by forming double-stranded DNA[22,23]. This alteration, in addition to steric effects, is likely a partial explanation for why addition of AO to HD22-7A-DAB restores thrombin activity to a level significantly higher than thrombin activity inhibited with free DAB. Testing this concept, we evaluated an antidote lacking two nucleotides that hybridize with the 5' terminus of the linker region. This AO (AO1) resulted in considerably reduced antidote efficacy. By contrast, truncation of the antidote's aptamer binding sequence by up to 20 nucleotides does not significantly reduce antidote efficacy (Fig. 2k). Further truncation of the AO resulted in decreased efficacy most likely because such short base pairing against the aptamer cannot reverse its binding. These results indicate that AOs that bind and constrain both the linker and at least part of the aptamer domain are optimal. With this insight, we developed an antidote for 11F7t-20A-DAB (AO5.4) that pairs with the entire linker region and part of the aptamer region; we observed that it effectively reverses the inhibition of this factor Xa EXACT inhibitor (Supplementary Fig. 6 and Supplementary Table 1).

## EXACT Inhibitor HD22-7A-DAB impedes multiple functions of thrombin

Thrombin has two exosites that mediate its enzymatic activity on multiple substrates, including fibrinogen, factors V, VIII, XI and XIII. We investigated how EXACT inhibitor HD22-7A-DAB impacts thrombin exosite-mediated cleavage of its natural substrates. Thrombin's cleavage of fibrinogen was characterized by a fibrin turbidity assay[24]. In the absence of an inhibitor, thrombin rapidly cleaves fibrinogen, leading to the formation of fibrin clots. Light scattering by the fibrin clots results in increased absorbance at 550 nm wavelength (Fig. 3a). DAB inhibits thrombin activity, resulting in a 7.7 min delayed onset of clot formation, a slower clot formation, and lower maximum absorbance, presumably due to altered clot structure (Fig. 3b). As expected, HD22 did not inhibit this thrombin activity, since it binds exosite II and does not block the fibrinogen-thrombin interaction or thrombin's active site[25]. In the presence of 50 nM EXACT inhibitor HD22-7A-DAB however, no significant absorbance increase was observed during the course of the experiment (120 min), indicating that thrombin's enzymatic activity on fibrinogen is completely inhibited. An equimolar mixture of HD22 and DAB did not yield a comparable potency to the EXACT inhibitor, highlighting the power of bimodal engineering and binding. To test the efficacy and kinetics of antidote reversal, AO2 was added to the thrombin-fibrinogen-HD22-7A-DAB mixture at the 10-min time point. After a 9.8-min lag time, the absorbance of the AO2-containing solution rapidly increased as clot formed (Fig. 3a, b). These results

demonstrate that the antidote can rapidly and effectively reverse the activity of the EXACT conjugate.

We then used SDS-PAGE analysis to characterize the effect of HD22-7A-DAB on FVIII activation which is mediated by thrombin ABE2 (Fig. 3c−e). Thrombin cleavage of FVIII results in the disappearance of both heavy and light chain of FVIII and appearance of several cleaved products (Fig. 3c, d). The presence of 100 nM DAB delayed such digestion. As expected, HD22 also showed an inhibitory effect due to its binding to ABE2. In the presence of HD22-7A-DAB, no proteolysis was observed for over 1 h, demonstrating a much higher potency than DAB and HD22 alone, and a similar potency to an equimolar mixture of HD22 and DAB (Fig. 3c and e).

## HD22-7A-DAB achieves potent, clinically relevant levels of anticoagulation

We then investigated the anticoagulation activity of HD22-7A-DAB in human plasma with a series of clinical clotting assays. Thrombin time (TT) directly probes the terminal step of the coagulation cascade, in which fibrinogen is cleaved by thrombin to form fibrin clots. As expected, free HD22 minimally affected TT as thrombin exosite II is not involved in fibrinogen cleavage (Fig. 4a). DAB, on the other hand, dose-dependently prolonged TT, resulting in a clotting time of 518 sec at the concentration of 250 nM. An equimolar mixture of HD22 and DAB prolonged TT to a similar extent as DAB alone. HD22-7A-DAB showed a different dose-dependent effect on TT compared to both DAB and free HD22; when the EXACT inhibitor concentration is lower than the added thrombin (13.4 nM), the conjugate showed limited effect on TT. However, once the EXACT inhibitor's concentration exceeded thrombin's concentration, TT dramatically increased and surpassed 999 s at a concentration of 31 nM. Notably, the high anticoagulant potency of HD22-7A-DAB can be largely reversed within 5 min by the addition of AO2.

The prothrombin time (PT) and activated partial thromboplastin time (aPTT) assays were used to assess the effect of HD22-7A-DAB on the extrinsic and intrinsic coagulation pathways, respectively (Fig. 4b, c). At a concentration of 500 nM, the EXACT inhibitor prolonged the PT and aPTT significantly longer than the same concentration of DAB. HD22 alone produced no anticoagulant effect, and equimolar mixtures of DAB and HD22 prolonged the PT and aPTT to an extent comparable to that of DAB alone. The anticoagulant effect of HD22-7A-DAB on both assays can also be efficiently reversed by AO2. However, HD22-7A-DAB could not completely block clotting in these assays because the EXACT inhibitor cannot prohibit activation of thrombin by factor Xa. Thus, HD22-7A-DAB will lose its effectiveness when the thrombin concentration eventually exceeds the concentration of the conjugate.

We further compared the anticoagulant effect of HD22-7A-DAB to that of unfractionated heparin (UFH), the most potent clinical anticoagulant. UFH activates the anticoagulant protein antithrombin, which then irreversibly inactivates multiple procoagulant proteases, particularly thrombin, factor Xa, and factor IXa, and results in high anticoagulant activity[26]. HD22-7A-DAB shows potency that rivals UFH in TT and PT assays (Supplementary Fig. 8), which can be attributed to its very high thrombin binding affinity. Not surprisingly, HD22-7A-DAB is a weaker anticoagulant than UFH in the aPTT assay, as HD22-7A-DAB does not inhibit factors IXa and Xa, two proteases that significantly affect aPTT.

Finally, we assessed the anticoagulation activity of HD22-7A-DAB in whole human blood with the activated clotting time (ACT), a point of care assay commonly utilized to monitor the level of anticoagulation in patients during cardiopulmonary bypass (CPB) and other invasive procedures that require rapid onset systemic anticoagulation[27]. HD22-7A-DAB dose dependently prolonged ACT (Fig. 4d). At 1 μM concentration, the ACT increased to 624 sec, significantly higher than the ACTs that followed the addition of DAB, HD22, or their equimolar

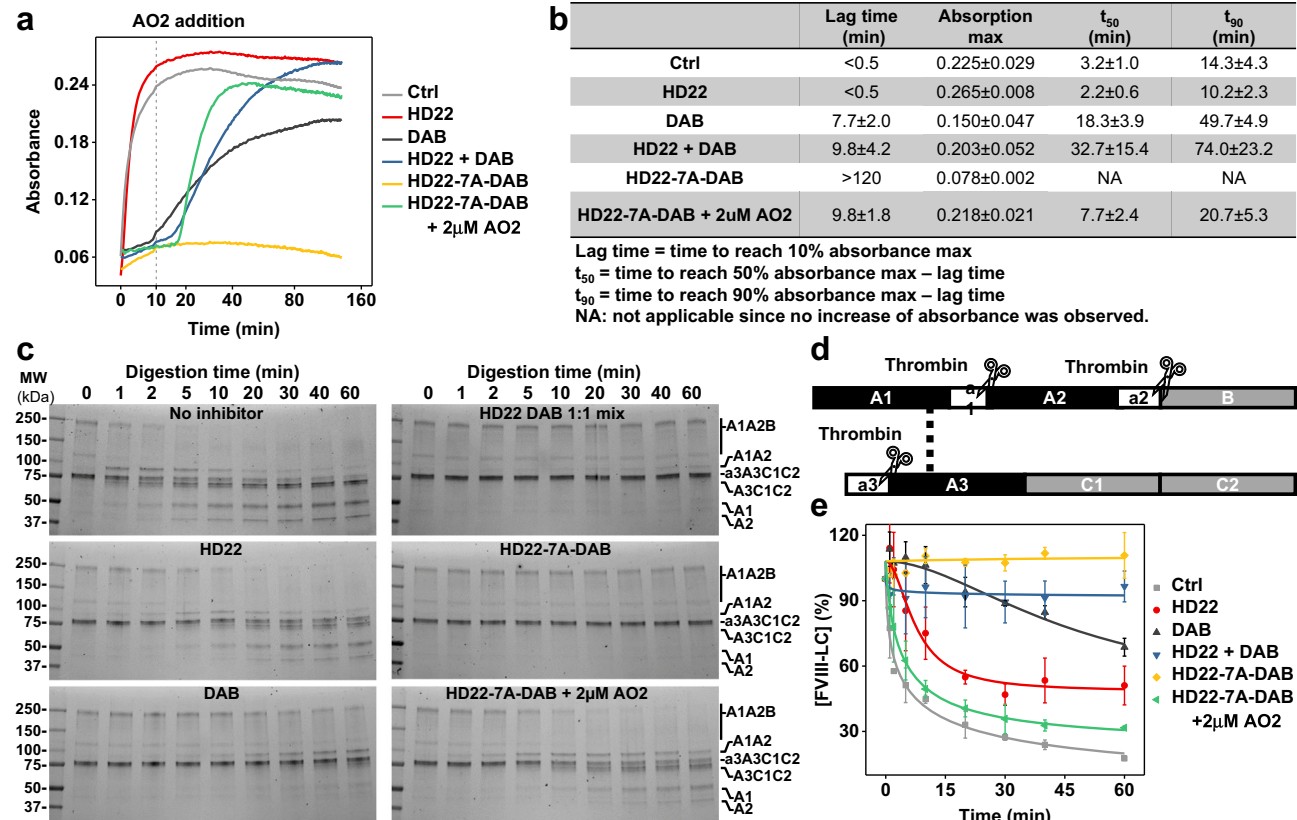

| | Lag time (min) | Absorption max | $t_{50}$ (min) | $t_{90}$ (min) |
|---|---|---|---|---|
| **Ctrl** | <0.5 | 0.225±0.029 | 3.2±1.0 | 14.3±4.3 |
| **HD22** | <0.5 | 0.265±0.008 | 2.2±0.6 | 10.2±2.3 |
| **DAB** | 7.7±2.0 | 0.150±0.047 | 18.3±3.9 | 49.7±4.9 |
| **HD22 + DAB** | 9.8±4.2 | 0.203±0.052 | 32.7±15.4 | 74.0±23.2 |
| **HD22-7A-DAB** | >120 | 0.078±0.002 | NA | NA |
| **HD22-7A-DAB + 2uM AO2** | 9.8±1.8 | 0.218±0.021 | 7.7±2.4 | 20.7±5.3 |

Lag time = time to reach 10% absorbance max
$t_{50}$ = time to reach 50% absorbance max – lag time
$t_{90}$ = time to reach 90% absorbance max – lag time
NA: not applicable since no increase of absorbance was observed.

**Fig. 3 | HD22-7A-DAB efficiently and reversibly inhibits multiple functions of thrombin. a, b** Inhibition of thrombin mediated fibrin clot formation by HD22-7A-DAB in a fibrinogen turbidity assay. **a** Time-course of fibrin clot formation with 0.8 mg/mL fibrinogen and 2.5 nM thrombin in the absence (gray) or presence of thrombin inhibitors (50 nM) including HD22 (red), DAB (black), equimolar mixture of HD22 and DAB mixture (blue), and EXACT inhibitor HD22-7A-DAB (yellow) characterized by the absorbance at 550 nm. The kinetics of antidote reversal was characterized by adding 2 μM of AO2 to the reaction 10 min after fibrinogen addition (green). **b** The lag time, maximum absorption, and time to reach 90% maximum absorption ($t_{90}$) in the absence and presence of different inhibitors were

determined. **c-d** Inhibition of thrombin (1 nM) mediated FVIII (100 nM) activation by EXACT inhibitor HD22-7A-DAB. **c** PAGE analysis of thrombin-mediated FVIII cleavage the absence or presence of thrombin inhibitors (100 nM), two sets of independent experiments were performed with similar results (n = 2). **d** Structure of human FVIII and thrombin cleavage site. **e** Thrombin's activation of FVIII in the absence and presence of thrombin inhibitors were quantified using the time-course concentration of intact light chain (a3A3C1C2) determined by band intensity. Data are presented as mean values +/– SD calculated from the two independent experiments.

mixture at the same concentration. More importantly, HD22-7A-DAB at 1 μM and 2 μM concentrations were able to prolong ACT to a level significantly higher than 5 U/mL UFH, the standard concentration employed during CPB. Those findings indicate an anticoagulant potency in whole blood that can potentially support highly thrombogenic procedures, including CPB. Thus HD22-7A-DAB is a groundbreaking aptamer-based, de novo generated EXACT inhibitor that can achieve such profound anticoagulant activity.

## Discussion

Small molecule active site inhibitors have proven incredibly clinically valuable to study and modulate enzymes involved in many human diseases; their use to target coagulation factors to treat and prevent cardiovascular disease and stroke has been particularly impactful. However, developing such inhibitors with both high target affinity and selectivity can be challenging given the low binding surface of the small molecules and the similarity among the active sites on different but related enzymes. Inspired by hirudin and other anticoagulants that target both EXosite and ACTive sites of key coagulation proteases, we show that this natural concept can be readily incorporated into rational drug design using chemically synthesized binding agents; such molecular engineering yields a generation of potent and rapidly reversible, EXACT inhibitors that distinguish between structurally homologous enzymes, such as thrombin and factor Xa. Once their

pharmacodynamic properties are optimized for various clinical applications, these potent and reversible anticoagulants may provide more effective and safer therapies to improve the acute care of millions of cardiovascular and stroke patients every year.

Strikingly, the binding properties of the active site-binding small molecule can be governed by the exosite-binding aptamer to which it is attached in three ways. First, the bimodal binding mechanism greatly increases the protease accessibility and potency of the active site-binding moiety. Second, the high binding synergy between the exosite-binding aptamer and the small molecule means that the aptamer's affinity for its exosite can dictate the binding selectivity of the small molecule while simultaneously enhancing it efficacy. Third, the reversal of active site inhibition by an antidote against the exosite-binding aptamer and the linker is particularly noteworthy as hemorrhage remains the chief safety issue associated with commonly utilized active site-targeted antithrombotic agents.

Aptamers in the past few decades have struggled to find their therapeutic niche and distinguish themselves from antibodies, small molecules, and other classes of therapeutics[28,29]. However, the ease with which oligonucleotides can be chemically conjugated to small molecules and the ability of aptamers to selectively bind surfaces of enzymes such as exosites[18–20,30–39] makes them particularly suitable for this type of rational drug design. Notably, although in HD22-7A-DAB the aptamer binds to the exosite II of thrombin, in theory, any aptamer

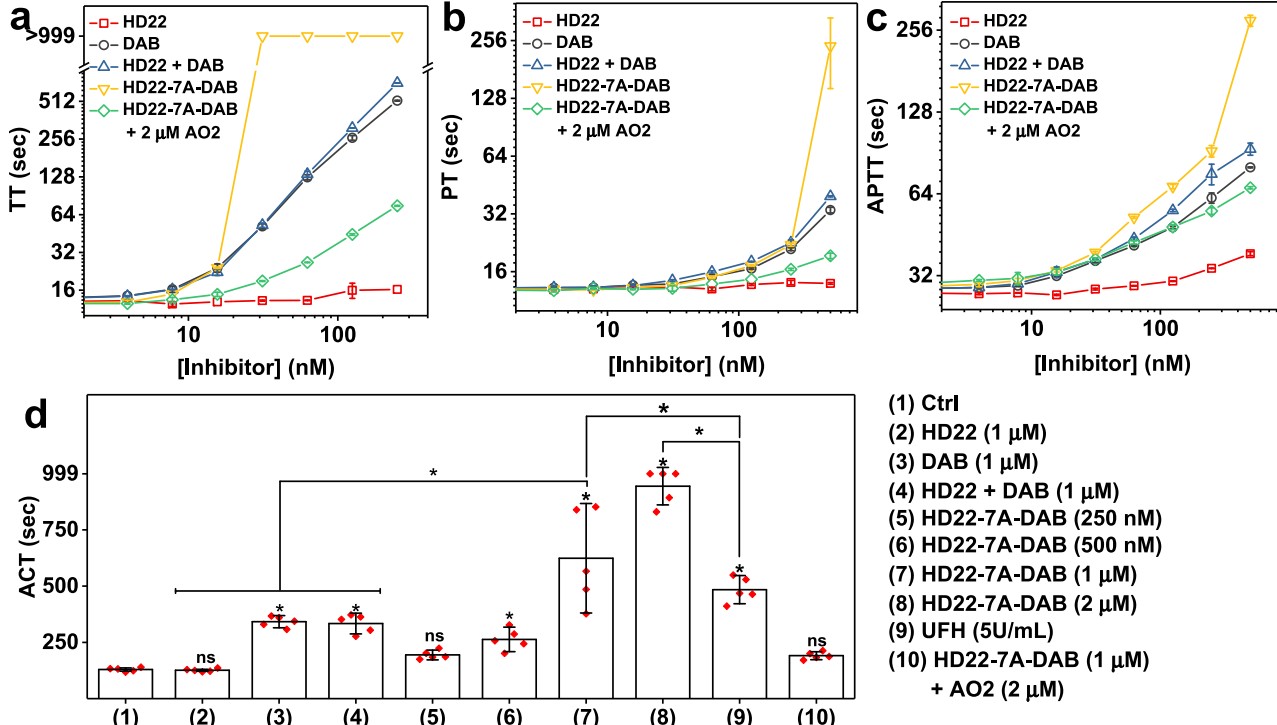

**Fig. 4 | Anticoagulation activity of EXACT inhibitor HD22-7A-DAB. a–c** Different concentrations of HD22, DAB, equimolar mixture of HD22 and DAB, or HD22-7A-DAB were added to normal human plasma and incubated for 5 min at 37 °C and (**a**) Thrombin time (TT), (**b**) Prothrombin time (PT), and (**c**) Activated partial thromboplastin time (aPTT) assays were performed to characterize anticoagulant efficacy. To characterize antidote reversal of HD22-7A-DAB, another 5 min incubation of antisense oligo AO2 (2 μM) with the plasma-HD22-7A-DAB mixture was performed. **d** Anticoagulant activity of HD22-7A-DAB in whole blood was characterized using the active clotting time (ACT) assay and compared to no inhibitor (Ctrl), HD22, DAB, equimolar mixture of HD22 and DAB, and UFH (unfractionated

heparin)-mediated anticoagulation. ACT exceeded the measurable range of the analyzer (999 sec) in the presence of 2 μM HD22-7A-DAB. In panels (**a–c**) data are presented as mean values +/– SD calculated from two independent experiments (*n* = 2) using pooled plasma. In panel (**d**) data are presented as mean values +/– SD calculated from blood collected from five individual donors. In penal (**d**) One-way ANOVA test followed by Bonferroni's multiple comparisons test (two-sided) was used to compare between two sets of data. *$P < 0.05$, ns, not significant. Exact $P$-values are provided as follows. (1) vs (2), (1) vs (5), and (1) vs (10): $p > 0.9999$; (1) vs (3): $p = 0.0002$; (1) vs (4): $p = 0.0003$; (1) vs (6): $p = 0.0418$; (7) vs (9): $p = 0.0288$; (1) vs (7), (1) vs (8), (1) vs (9), (2) vs (7), (3) vs (7), (4) vs (7), and (8) vs (9): $p < 0.0001$.

which binds outside of enzyme's catalytical center can be utilized to develop an effective EXACT inhibitor. The aptamer's binding affinity and linker govern the synergy with the small molecule and result in strong active site inhibition. Therefore, potent, selective, and reversible EXACT inhibitors can be created for virtually any enzyme, with or without defined exosites, by appropriately linking a high affinity, high specificity aptamer to even a low affinity, low specificity, small molecule active site inhibitor. For enzymes with functionally important exosites however, we anticipate that the most effective EXACT inhibitors will contain aptamers that both inhibit exosite function as well as catalytic activity. These studies foreshadow the development of a large class of highly potent and specific enzyme inhibitors that are rapidly reversible. This combination of beneficial properties has been difficult to obtain for any therapeutic agent but now appears to be straight forward to attain. Moreover, this generalizable molecular engineering approach can help resuscitate many small molecule drugs abandoned because they were not effective or selective enough on their own.

## Methods

### Materials
All DNA oligos were purchased from Integrated DNA Technologies Inc. Modified RNA oligos were synthesized in house using a Mermade oligo synthesizer followed by HPLC purification. Chemicals were purchased from Sigma without specification. Dabigatran (DAB) and apixaban-COOH (APX) were purchased from AK Scientific Inc. Human alpha thrombin, human factor Xa, and human fibrinogen were purchased from Haematologic Technologies Inc. Recombinant human factor VIII

was purified from Kogenate FS (Bayer) and quantified by UV absorbance. Pooled normal human plasma was purchased from George King Bio-Medical. Human whole blood samples in this work were obtained under Duke IRB protocol (Pro00007265).

### Synthesis and purification of EXACT inhibitors with poly (A) linker
EXACT inhibitors were synthesized via EDC/NHS using a reported method with modification[40]. Briefly, 1 μmole of EDC and 4 μmole of NHS were mixed in 22 μL of $H_2O$/DMSO solution (v:v = 15%:85%). The mixture was immediately added to 230 μL of 5.5 mM DAB or APX dissolved in DMSO. After 30 min of incubation at room temperature, 5 nmole of amine-modified aptamer derivatives dissolved in 60 μL of 420 mM TEA/HCl buffer, pH 10 were added into the mixture and incubated at room temperature overnight. The unreacted small molecules were removed by ethanol precipitation, and the conjugates were purified from the unreacted aptamer and other residue impurities by HPLC and verified by IES-MS. The purity of the final product was validated on a 15% denaturing PAGE (Supplementary Fig. 1).

### Fluorogenic activity assay
All reagents were freshly reconstituted in the reaction buffer (20 mM HEPES, 150 mM NaCl, and 2 mM $CaCl_2$, 0.02%, Tween 20, pH 7.5) before the assay. 10 μL of thrombin or factor Xa (final concentration 0.5 nM) were first incubated with 5 μL of inhibitor, inhibitor/antidote mixture, or buffer control on a 384-well opaque plate for 5 min at 28 °C. 10 μL of fluorogenic substrate (final concentration 50 μM) was

then added to the mixture and the time-course fluorescence ($\lambda_{ex} = 352$ nm, $\lambda_{em} = 470$ nm) of the sample were recorded every minute for 15 min at 28 °C using a SpectraMax i3 microplate reader (Molecular Devices). The catalytic rate of the protease was quantified by the slope from linear regression of the time-dependent fluorescence intensity and normalized with the sample containing no inhibitor as 100%. Each experiment was performed in triplicates.

## Bio-layer interferometry

Bio-layer interferometry was performed using Octet R8 BLI System (Sartorius). Briefly, 3.3 nM of biotinylated HD22-7A-DAB, HD22-7A-NH₂, HD23-7A-DAB, and HD23-7A-NH₂ were immobilized to Octet streptavidin biosensors (Sartorius). The baseline was collected in buffer (20 mM HEPES, 150 mM NaCl, and 2 mM CaCl2, 100x BSA, pH 7.5) followed by a 30 min association in buffer containing different concentrations of thrombin. A 30 min dissociation was performed in buffer containing 200 nM HD22-7A-DAB. The existence of free HD22-7A-DAB in dissociation buffer prevents re-binding of thrombin and provides more accurate $k_{off}$ measurements.

## Size exclusion chromatography

All experiments were performed using a BioCADRPM perfusion chromatography workstation with a HiLoad 16/600 Superdex 200 pg column (cytiva). Before experiment, the column was equilibrated overnight in buffer (20 mM HEPES, 150 mM NaCl, and 2 mM CaCl2, 0.1%, PEG8000, pH 7.5). 1.25 nmole of thrombin, HD22-7A-DAB, or their equimolar mixture in 250 µL buffer was loaded to the column with a buffer elution rate of 0.5 mL/min for at least 240 min. The absorbance at 260 nm was recorded throughout the experiment. The protein standard consisted of human IgG (150KDa), BSA (67KDa), thrombin S195A (38KDa), and a nanobody (14KDa).

## X-ray crystallography

A mixture of 150 mM IIa$_{S195A}$ and 160 mM HD22-7A-DAB in 20 mM HEPES, 0.15 M NaCl, pH 7.4 was mixed with an equal volume of 0.1 M MES monohydrate, pH 6.0, 22% (v/v) polyethylene glycol 400 and crystals were grown from 2 µl sitting drops by vapor diffusion. X-ray diffraction data were collected at beamline 17-ID-1 (AMX) at NSLS-II. Data were merged and scaled using the AutoProc pipeline[41] using XDS[42], Aimless[43], Pointless[44] and StarAniso. Molecular replacement was done using the human thrombin structure 1PPB[45] and the structure of HD22 from 5EW1[46] using Phenix.Phaser[47]. Initial rounds of model completion and refinement were done with COOT[48] and Phenix.Refine[47]. The last round of refinement was done using PDBREDO[49].

## Fibrinogen turbidity assay

All reagents were freshly reconstituted in the reaction buffer the assay. 20 µL Thrombin (final concentration 2.5 nM) was incubated with 10 µL thrombin inhibitors (final concentration 50 nM) or buffer control at 37 °C for 5 min. 20 µL fibrinogen (final concentration 0.8 mg/mL) was then added to the reaction and absorbance at 550 nm was measured every 30 s over 130 min using a SpectraMax i3 microplate reader to monitor clot formation. To test the kinetics of antidote reversal, 1 µL AO2 (final concentration 2 µM) was added to the reaction 10 min after fibrinogen addition. The maximum absorption over the period of assay in the absence and presence of different inhibitors were determined. The time to reach 10%, 50%, and 90% maximum absorption increase from the start point were calculated and recorded as lag time, $t_{50}$, and $t_{90}$, respectively, for each inhibitor to evaluate the kinetic of fibrinogen activation. Each experiment was performed in triplicates to determine the mean and standard deviation of each parameter.

## Thrombin cleavage of FVIII

All reagents were freshly reconstituted in the reaction buffer the assay. Thrombin (final concentration 1 nM) was incubated in the absence or presence of thrombin inhibitors (final concentration 100 nM) at 37 °C for 5 min following by addition of recombinant human FVIII (final concentration 100 nM). Sample were collected at 1, 2, 5, 10, 20, 30, 40, and 60 min of reaction and quenched in SDS loading buffer and heating at 95 °C for 5 min. The digestion products at different time points are then characterized on a 4–20% PAGE gel. Thrombin's activity on activating FVIII in the absence and presence of thrombin inhibitors were quantified using the time-course concentration of intact light chain under thrombin digestion determined by band intensity. The assay was performed in duplicates.

## Plasma coagulation assays

Thrombin time (TT), prothrombin time (PT), and activated partial thromboplastin time (aPTT) assays were performed in citrated normal human plasma on a hemostasis coagulation analyzer (Diagnostica Stago). For TT, 5 µL of inhibitors in the reaction buffer were mixed with 100 µL of plasma and incubated at 37 °C for 5 min. 50 µL of thrombin (6 NIH units/ml) in the reaction buffer was then added to initiate clotting. For PT, 5 µl of inhibitors in the reaction buffer were mixed with 50 µL of plasma and incubated at 37 °C for 5 min. 100 µL of TriniCLOT PT Excel S reagent was then added to initiate clotting. For aPTT, 5 µL of inhibitors in the reaction buffer were mixed with 50 µL of plasma and incubated at 37 °C for 5 min, 50 µL of TriniCLOT aPTT S reagents was then added followed by another 5 min incubation at 37 °C. Finally, 50 µL of 20 mM CaCl₂ was added to initiate clotting. To characterize antidote reversal of HD22-7A-DAB in the above assays, 5 µL AO2 (finial concentration 2 µM) in the reaction buffer was added after 5 min incubation between plasma and HD22-7A-DAB, followed by another 5 min of incubation before the next step. All assays are performed induplicates.

## Active clotting time (ACT)

Citrated blood (72 µL) freshly collected from heathy donors were incubated with 6 µL inhibitors reconstituted in the reaction buffer at room temperature for 3 min following addition of 2.1 µL CaCl₂ (245 mM). The blood mixture was then immediately analyzed on an ACT+ cuvette (Accriva Diagnostics) using a Hemochron Jr Signature (Instrumentation Laboratory). The assay was performed with a N value of five. One-way ANOVA test was used to compare between two sets of data.

## Statistical analysis

For multiple comparison within a data set, we performed ordinary one-way ANOVA followed by Bonferroni test between every two groups. A significant difference was called when $P$-value $< 0.05$.

## Reporting summary

Further information on research design is available in the Nature Portfolio Reporting Summary linked to this article.

## Data availability

Crystallographic data for the structures reported in this article have been deposited to the protein data bank with PDB ID 8TQS. The authors declare that other data supporting the findings of this study are available within the paper and its Supplementary Information files. Source data are provided with this paper and accessible via Figshare: [https://doi.org/10.6084/m9.figshare.25320667]. Source data are provided with this paper.

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

## Acknowledgements

We thank Dr. Dario Neri for technical assistance on oligo conjugation; Dr. Yi Xiao for useful discussion on binding characterization. This work was supported by a US National Institutes of Health grant (P01-HL139420) to S. Krishnaswamy and B.A.S. and AHA postdoctoral fellowship (23POST1018721) to H.Y.

## Author contributions

Experiments were designed and conceived by H.Y., S.Kumar, S. Krishnaswamy, and B.A.S. Experiments were performed by: H.Y., S.Kumar. Collection of blood reagents was supported by G.P., R.R. and S.F. Synthesis and purification of oligonucleotides was supported by A.Y. Data were analyzed by H.Y., S. Kumar, J.W.F., V.N.K., G.P., J.L., R.R., S. Krishnaswamy and B.A.S. H.Y. and B.A.S. conceived the idea and wrote the manuscript. B.A.S. and S. Krishnaswamy acquired the funding, provided resources and supervision. All of the authors contributed to editing of the manuscript and support the conclusions.

## Competing interests

Duke has submitted a patent application (U.S. Provisional Application No. 63/421,756) on the thrombin and factor Xa EXACT inhibitors. H.Y. and B.A.S. are inventors on such Duke Intellectual Property. The remaining authors declare no competing interests.

## Ethics

This research complies with ethical regulations under Duke University Health System (DUHS) Institutional Review Board for Clinical Investigations with Federal wide Assurance No: FWA 00009025
