## [Peer Review File · Nature Communications]

Aptameric hirudins: engineering potent, selective, and reversible EXosite-ACTive site (EXACT) inhibitorsREVIEWER COMMENTS

Reviewer #1 (Remarks to the Author):

To the Authors,

The manuscript "Aptameric hirudins: engineering potent, selective, and reversible EXosite-ACTive site (EXACT) inhibitors by Yu et al. is a tremendously interesting and exciting body of work. This study demonstrates that exosite aptamers are essential in enhancing the anticoagulant ability of active site inhibitors, producing a potent anticoagulant therapeutic. Additionally, reversing the anticoagulant activity of these inhibitors by antisense oligonucleotides is vital for using these inhibitors as therapeutics for cardiovascular surgeries. Given the importance of generating new and improved anticoagulants with corresponding antidotes, this novel approach warrants publication in Nature Communications. This work is novel and advances the field. I do have a few specific comments/concerns that could be clarified by the authors, as outlined below:

- How important is it that the EXACT inhibitor binds to the exosite? Can any high-affinity binding aptamer that binds to other sites (outside the exosite) on the protein exert the same anticoagulant effect?
- Do heparin or other glycosaminoglycans affect this inhibition? Does heparin compete with HD22 for binding to thrombin's exosite 2?
- While the EXACT inhibitors' kinetics are mentioned, I am interested in knowing the on-and-off rate of HD22 and DAB. Do the bivalent molecules increase the off rate of HD22 while decreasing the on rate of DAB? As shown in Figure 1C, when HD22 and DAB are incubated together at lower concentrations, the inhibition resembles HD22, while at higher concentrations of thrombin, the inhibition resembles DAB. While the K_d for HD22-7A-NH₂ and HD23-7A-DAB support the model, it would have been nice to see the K_d s for the "free" (HD22 and DAB) since they are assessed in Figure 1c.
- Are there any conformational changes in the molecule upon binding the HD22-7A-DAB bivalent inhibitor?
- Does the spacer stabilize (in some way) the binding of HD22-7A-DAB?
- The 11F7t-20A-DAB EXACT inhibitor is fascinating, as connecting DAB to a high-affinity factor Xa aptamer results in a significantly better inhibitor of factor Xa. Can the authors explain or at least speculate further on this fascinating result?
- What is the effect of generating an AO to just the linker region? Is that possible?
- In Figure 3c, the gel for HD22 DAB 1:1 and HD22-7A-DAB looks similar; however, the authors say on page 5, line 26, that there is a... higher potency than DAB, HD22, or their equimolar mixture.

Reviewer #2 (Remarks to the Author):

The authors describe a strategy for creating potent and selective anticoagulation agents by linking exosite-targeting aptamers to small molecule active site inhibitors of thrombin and factor Xa. The resulting inhibitors are called exosite-active site (EXACT) inhibitors. They solved the crystal structure of an EXACT inhibitor, HD22-7A-DAB, bound to thrombin, showing unambiguous binding of the aptamer and the small molecule inhibitor to predicted regions of the protein surface. They present clear data demonstrating the inhibition of thrombin activity and its superiority to the individual reagents. The authors also present promising data of HD22-7A-DAB inhibiting different functions of thrombin and anticoagulation measured by the active clotting time. Overall, the work advances the field of therapeutic aptamers and synthetic biology, as well as knowledge of thrombin and proteases, and should be of interest to the readers of Nature Communications.

Minor comments:

1. It is unclear how antidote oligonucleotides (AOs) inhibit EXACT molecules. Is the aptamer antisense sequence necessary (i.e., does the AO bind to the aptamer?) or is a poly-T oligonucleotide sufficient to hinder EXACT binding to its target via base pairing to the polyA region? Have the authors performed control experiments with a poly-T oligonucleotide or an AO with a scrambled 'aptamer' antisense sequence?
2. In the functional assays, AOs are 20 – 40 fold more concentrated than the EXACT inhibitors. Have the authors performed AO titration experiments and can the authors comment on the implications of high AO requirements in the clinical setting?
3. The current conclusion is a concise summary of the manuscript. However, some readers may be interested in learning more about the broader implications of their findings (i.e., provide a perspective of how EXACT thrombin/factor Xa inhibitors contribute to the field, what can be improved such as optimizing AOs, etc).
4. "...crystals were grown from 2 ml sitting drops by vapor diffusion" appears to be a typo from microliters.

Point-to-point response to reviewer's comments

We thank the reviewers for their careful reading of our manuscript and their many constructive comments and thoughtful suggestions. We have incorporated several suggested changes into the revised manuscript which we believe is significantly improved. Please find following a detailed response to your comments and suggestions along with three data figures that support wording and figure changes that we have incorporated into the paper as you suggested.

Reviewer #1:

1. How important is it that the EXACT inhibitor binds to the exosite? Can any high-affinity binding aptamer that binds to other sites (outside the exosite) on the protein exert the same anticoagulant effect?

We thank the reviewer for this important question. Because synergy is largely obtained by bivalent binding, the aptamer should enhance the potency and specificity of the small molecule active site inhibitor even if the aptamer binds the target enzyme outside of an exosite. The ability of aptamer HD22 binding to exosite II may further enhance the potency of the inhibitor as it can independently inhibit additional functions of thrombin. However, since the thrombin cleavage of its tripeptide substrate does not involve exosite-binding, we believe that aptamer binding to a specific exosite is not essential to achieve high potency for EXACT inhibitors. Therefore, we anticipate that EXACT inhibitors can even be developed for enzymes without functionally defined exosites. Nevertheless, for enzymes with functional exosites, we anticipate that the most effective EXACT inhibitors will bind both the exosite and active site on the protein and inhibit the function of both sites. We have expanded the discussion to make this important point.

2. Do heparin or other glycosaminoglycans affect this inhibition? Does heparin compete with HD22 for binding to thrombin's exosite 2?

We thank the reviewer for this question. Previous studies have shown that HD22 competes with heparin for binding to thrombin (*J Mol Biol.* 1997:688-98.). We therefore performed thrombin-based fluorogenic

cleavage assays to compare inhibition curves with HD22-7A-DAB in the absence or presence of 5U/mL unfractionated heparin (UFH). Interestingly, UFH at this high dose was unable to compete with HD22-7A-DAB in this functional assay(Fig. R1). We attribute this inability to compete to the ultra-high thrombin affinity of the EXACT inhibitor compared to the low thrombin binding affinity of heparin ($K_D \sim \mu\text{M}$, *J Biol Chem.* 266:6342-52). Notably, the difference between the two thrombin cleavage inhibition curves at lower EXACT inhibitor concentrations is because UFH itself reduces the thrombin cleavage rate by $\sim 20\%$ in this experiment.

Figure R1. HD22-7A-DAB mediated inhibition of thrombin activity in the absence and presence of heparin (UFH).

3. While the EXACT inhibitors' kinetics are mentioned, I am interested in knowing the on-and-off rate of HD22 and DAB. Do the bivalent molecules increase the off rate of HD22 while decreasing the on rate of DAB? As shown in Figure 1C, when HD22 and DAB are incubated together at lower concentrations, the inhibition resembles HD22, while at higher concentrations of thrombin, the inhibition resembles DAB. While the K_d for HD22-7A-NH2 and HD23-7A-DAB support the model, it would have been nice to see the K_d s for the "free" (HD22 and DAB) since they are assessed in Figure 1c.

We thank the reviewer for this question. As presented in Fig. S5, we did observe that the bivalent HD22-7A-DAB demonstrated an increased k_{on} and a decreased k_{off} compared with both monovalent ligands. Regarding Figure 1C, the biphasic inhibition curve of HD22 and DAB mixture likely results because the aptamer has a higher affinity but lower inhibition max for thrombin, while DAB has lower affinity but higher inhibition max. Thus, the aptamer dominates the effect at lower concentrations while the DAB dominates at higher concentrations. We agree that it would be nice to see the K_{DS} for the free HD22 and DAB, but unfortunately, the surface-dependent nature of BLI does not allow us to quantify these K_{DS} of directly. However, in the thrombin cleavage fluorogenic assay, we observed that A.) free HD22 and HD22-7A-NH₂ show no significant difference in inhibitory activity, and similarly B.) DAB and HD23-7A-DAB did not show no significant difference (Fig S4). These results strongly suggest that the K_{DS} of free HD22 and DAB are similar to those of HD22-7A-NH₂ and HD23-7A-DAB and we do report these numbers in Fig. S5.

4. Are there any conformational changes in the molecule upon binding the HD22-7A-DAB bivalent inhibitor?

We thank the reviewer for this important question. Our X-ray crystallography data does not reveal any obvious conformational changes in thrombin upon HD22-7A-DAB binding, the structure of the aptamer domain also overlays well with previously reported thrombin-bound HD22. We have added this important point to the text in the revised manuscript.

5. Does the spacer stabilize (in some way) the binding of HD22-7A-DAB?

We thank the reviewer for this question. Our results indicate that the spacer is essential to inducing synergistic binding, which increases the binding affinity of the EXACT inhibitor compared to monovalent

ligands. Thus, from this perspective, the linker does stabilize the binding. However, we did not find evidence that the linker directly interacts with the surface of thrombin to stabilize binding.

6. The 11F7t-20A-DAB EXACT inhibitor is fascinating, as connecting DAB to a high-affinity factor Xa aptamer results in a significantly better inhibitor of factor Xa. Can the authors explain or at least speculate further on this fascinating result?

We thank the reviewer for this suggestion and have elaborated on this interesting observation in the discussion of the revised manuscript.

7. What is the effect of generating an AO to just the linker region? Is that possible?

We thank the reviewer for this question. We have now tested several shorter AOs of HD22-7A-DAB (**Fig. R2**, revised Fig. 2j,k). The result revealed that truncation of AO down to 16 nt (AO7) did not show much difference from the full-length AO (AO2) at reversing EXACT inhibitor activity. This reiterated that only a portion of the aptamer needs to be hybridized for optimal reversibility. However, further shortening of AO to 12 (AO8) or 7 nt (AO9) dramatically reduces its efficacy. Clearly, for this EXACT inhibitor with a 7nt linker, an AO that only hybridizes with the linker is not effective. To understand if an EXACT inhibitor with a longer linker can be reversed by an AO that only hybridizes with the linker. We performed similar experiments with HD22-20A-DAB (**Fig. R3**). We observed that the anti-linker AO (AO11) does have a partial reversal effect, presumably resulting from a reduction in synergy, steric hindrance, or both. However, to show optimal reversibility, both the linker and at least a portion of the aptamer need to form base pairs with the antidote oligo. We have added useful information to the revised figure 2 and the discussion in the revised manuscript.

Figure R2. Revised fig.2 j,k

Figure R3. Reversal of HD22-20A-DAB by different antidotes

8. In Figure 3c, the gel for HD22 DAB 1:1 and HD22-7A-DAB looks similar; however, the authors say on page 5, line 26, that there is a... higher potency than DAB, HD22, or their equimolar mixture.

We thank the reviewer for this comment and correction. We agree with the reviewer and modified this statement appropriately. It now reads: "... higher potency than DAB and HD22 alone, and a similar potency to an equimolar mixture of HD22 and DAB."

Reviewer #2:

1. It is unclear how antidote oligonucleotides (AOs) inhibit EXACT molecules. Is the aptamer antisense sequence necessary (i.e., does the AO bind to the aptamer?) or is a poly-T oligonucleotide sufficient to hinder EXACT binding to its target via base pairing to the polyA region? Have the authors performed control experiments with a poly-T oligonucleotide or an AO with a scrambled 'aptamer' antisense sequence?

We thank the reviewer for this comment. Please see our response to comment #7 of Reviewer #1 above and as note we have added additional discussion about this interesting point and modified figure 2 in the revised manuscript with the additional data provided above.

2. In the functional assays, AOs are 20 – 40 fold more concentrated than the EXACT inhibitors. Have the authors performed AO titration experiments and can the authors comment on the implications of high AO requirements in the clinical setting?

We thank the reviewer for this question. We agree with the reviewer that a much lower dose of AO may be sufficient in some assays and plan to optimize AO concentration in a follow-up study. However, we chose the AO concentration of 2 μ M in these studies because different EXACT inhibitor concentrations are used in different functional assays. In the fluorogenic assays, the EXACT inhibitor concentration is up to 250 nM; in PT and aPTT assays, the inhibitor concentration is up to 500 nM; in ACT assay, the inhibitor concentration is 1 μ M. For consistency, we decided to utilize the same AO concentration to evaluate the efficacy of AO in all these assays; thus 2 μ M was chosen as it is 2X the highest concentration of EXACT inhibitor utilized (in the ACT assay). In the future, additional optimization of the AOs will be performed before translating it into animal thrombosis and human ex vivo blood circulating surgical models. Such optimization will include exploring the use of nucleotide and backbone modifications on the AOs to improve their affinities for the EXACT inhibitor and further reduce the excess of AO required to reverse the EXACT inhibitor's anticoagulant activity.

3. *The current conclusion is a concise summary of the manuscript. However, some readers may be interested in learning more about the broader implications of their findings (i.e., provide a perspective of how EXACT thrombin/factor Xa inhibitors contribute to the field, what can be improved such as optimizing AOs, etc).*

We thank the reviewer for this suggestion and comment. We have expanded the papers discussion in the revised manuscript to provide the reader some thoughts about the broader implications of this work.

4. *“...crystals were grown from 2 ml sitting drops by vapor diffusion” appears to be a typo from microliters.*

We thank the reviewer for catching this typo. We have made the appropriate correction in the revised text.

REVIEWERS' COMMENTS

Reviewer #1 (Remarks to the Author):

All of my concerns regarding this manuscript has been sufficiently answered by the authors. Also, reading the revised manuscript, I am happy with the additions, which adds clarity to the work. I feel that the work is important and should be published. The results are novel and is significant in the field. Thus, I do not have any additional concerns.

Reviewer #2 (Remarks to the Author):

I am satisfied with the author's changes to the manuscript.

Point-to-point response to reviewer's comments

We thank the reviewers for their careful reading of our revised manuscript and appreciate their comments about it addressing all of their original concerns and should be now published.